# Prognostic Significance of the Extranodal Extension of Regional Lymph Nodes in Stage III-N2 Non-Small-Cell Lung Cancer after Curative Resection

**DOI:** 10.3390/jcm10153324

**Published:** 2021-07-28

**Authors:** Beatrice Chia-Hui Shih, Jae Hyun Jeon, Jin-Haeng Chung, Hyun Jung Kwon, Jeong Hoon Lee, Woohyun Jung, Yoohwa Hwang, Sukki Cho, Kwhanmien Kim, Sanghoon Jheon

**Affiliations:** 1Department of Thoracic and Cardiovascular Surgery, Seoul National University Bundang Hospital, Seongnam-si 13620, Korea; beatriceshih@snubh.org (B.C.-H.S.); chucky0406@gmail.com (W.J.); yooflower@snubh.org (Y.H.); skcho@snubh.org (S.C.); kmkim0070@snubh.org (K.K.); jheon@snu.ac.kr (S.J.); 2Department of Thoracic and Cardiovascular Surgery, Seoul National University College of Medicine, Seoul 03080, Korea; 3Department of Translational Medicine, Seoul National University Bundang Hospital, Seongnam-si 13620, Korea; jhchung@snubh.org (J.-H.C.); treeish@naver.com (H.J.K.); 4Department of Pathology, Seoul National University College of Medicine, Seoul 03080, Korea; 5Division of Biomedical Informatics, Seoul National University Biomedical Informatics, Seoul National University College of Medicine, Seoul 03080, Korea; sosal@snu.ac.kr

**Keywords:** lung cancer, prognosis, lung pathology, extranodal extension, lung cancer surgery, lymph nodes

## Abstract

The present study investigated the prognostic role of extranodal extension (ENE) in stage III-N2 non-small-cell lung cancer (NSCLC) following curative surgery. From January 2005 to December 2018, pathologic stage III-N2 disease was diagnosed in 371 patients, all of whom underwent anatomic pulmonary resection accompanied by mediastinal lymph node dissection. This study included 282 patients, after excluding 89 patients who received preoperative chemotherapy or incomplete surgical resection. Their lymph nodes were processed; after hematoxylin and eosin staining, histopathologic slides of the metastatic nodes were reviewed by a designated pathologist. Predictors of disease free survival (DFS), including age, sex, operation type, pathologic T stage, nodal status, visceral pleural invasion, perioperative treatment, and the presence of ENE, were investigated. Among the 282 patients, ENE was detected in 85 patients (30.1%). ENE presence was associated with advanced T stage (*p* = 0.034), N2 subgroups (*p* < 0.001), lymphatic invasion (*p* = 0.001), and pneumonectomy (*p* = 0.002). The multivariable analysis demonstrated that old age (*p* < 0.001), advanced T stage (*p* = 0.012), N2 subgroups (*p* = 0.005), and ENE presence (*p* = 0.005) were significant independent predictors of DFS. The DFS rate at five years was 21.4% in patients who had ENE and 43.4% in patients who did not have ENE (*p* < 0.001). The presence of ENE, coupled with tumor-node-metastasis staging, should be recognized as a meaningful prognostic factor in stage III-N2 NSCLC patients.

## 1. Introduction

Lung cancer continues to be the top global cause of cancer-related mortality. In particular, N2 stage III non-small-cell lung cancer (NSCLC) patients continue to have a bleak prognosis, owing to the high incidence of metastasis or tumor recurrence [1]. Although multimodal treatments, including curative resection, perioperative chemotherapy and radiotherapy, have been implemented to enhance the overall outcomes, patients with stage III-pN2 NSCLC have unfavorable survival, with a rate reported to be 24% in an analysis of cases from the the Surveillance, Epidemiology, and End Results (SEER) dataset [2]. Despite considerable efforts to identify new treatment methods and prognostic predictors, treatment guidelines continue to be based on the tumor-node-metastasis (TNM) classification of the cancer.

For stage III-N2 NSCLC patients, the eighth edition of the TNM classification recommends a detailed categorization of the N component of staging, distinguishing N2 single-station involvement without N1 metastasis (N2a1), N2 non-skip single-station metastasis (N2a2), and N2 multistation disease (N2b), and there is significant survival variability among these heterogeneous groups. The survival rate at five years for patients with bulky N2b is 5–8%, whereas it is 35% for those with N2a1 [3]. Given the heterogeneity of the N2 lung cancer population and their poor prognosis despite different perioperative therapies [4,5], identifying high risk factors for tumor recurrence within this subgroup will help to predict patients’ long term prognosis and to select suitable strategies for treatment. Unlike solid cancers (e.g., breast, thyroid, colorectal, or prostate cancers), where the characteristics of lymph node (LN) involvement are well defined, the prognostic power of detailed aspects of LN involvement has not yet been fully delineated for lung cancer.

Extranodal extension (ENE) of involved LNs, which is characterized by the presence of cancer cells extending through the LN capsule into the surrounding fibrous adipose tissue, is already a well recognized prognostic factor in solid tumors, such as head/neck, breast, pancreas, prostate, and colorectal cancers [6,7,8,9,10,11,12,13,14,15,16,17,18,19]. However, few recent studies have reported data regarding the importance of ENE in NSCLC patients [20,21]. Moreover, ENE status is not included as a prognostic factor in the current TNM system [22].

Hence, the present study investigated the clinicopathologic characteristics related to stage III-N2 patients with ENE and their prognosis according to N2 subgroups after curative pulmonary lung resection.

## 2. Materials and Methods

### 2.1. Patient Selection

This study received approval from the Seoul National University Bundang Hospital Institutional Review Board (IRB number: B-2012/664-105). A waiver for patient consent was granted due to the retrospective design of the study. A medical record review was retrospectively conducted for 371 patients with stage III-N2 NSCLC who underwent anatomic lung resection with systemic LN evaluation from January 2005 to December 2018, according to the STROBE (strengthening the reporting of observational studies in epidemiology) statement (www.strobe-statement.org, accessed on 26 July 2021) [23]. A STROBE checklist can be found in the Appendix A. The following exclusion criteria were applied: (1) incomplete resection (*n* = 22) and (2) neoadjuvant therapy (*n* = 67). After excluding these 89 patients, the analysis included 282 patients. Clinicopathologic features, including age, sex, smoking history, preoperative lung function, pathologic TNM stage, histologic cell type, extent of lung resection, histopathologic features, adjuvant treatment and the presence of ENE, were analyzed. The clinicopathologic features related to ENE and their influence on postoperative survival and recurrence after surgery with curative intent were investigated.

### 2.2. Histopathologic Review

In the study, all the patients underwent complete anatomic pulmonary resection and mediastinal LN dissection. The histologic slides were analyzed by a dedicated pathologist (J-H Chung). The specimens were stained with hematoxylin and eosin. The definition of ENE used in the present study was extension of cancer cells through the LN capsule and invasion into surrounding fibrous adipose tissue (Figure 1).

### 2.3. Follow Up

Follow up after surgery was performed at 3-month intervals for the initial 24 months. Subsequently, follow-up was performed biannually for next 3 years. After the fifth year, the surveillance was performed annually. During follow up, contrast enhanced computed chest tomography was performed according to the schedule. Recurrence was defined as histologic confirmation or radiologic findings of a tumor. The recurrence date was documented as the date of the first examination detecting recurrence. Disease free survival (DFS), which was the primary study outcome, was documented as the interval between surgical resection and either recurrence or death. Recurrence patterns were classified as distant metastasis or locoregional recurrence. The definition of distant recurrence was the presence of a recurrent tumor observed at a site different from that of the treated primary tumor and ipsilateral pulmonary recurrence that did not meet the aforementioned criteria. Locoregional recurrence was defined as observed recurrence at an anatomically contiguous site from the operative site—for instance, the bronchial resection margin or lung resection borders—and at the regional LNs of the primary tumor or pleural seeding. The definition of distant recurrence was the presence of a recurrent tumor observed at a site different from that of the treated primary tumor and ipsilateral pulmonary recurrence that did not meet the aforementioned criteria.

### 2.4. Statistical Analysis

To compare variables across the unmatched groups, descriptive statistics such as Student t-test for quantitative variables and the Pearson’s chi-square or Fisher’s exact tests for qualitative variables were used. The Kaplan–Meier curve was plotted to estimate DFS, and the log rank analysis was performed to identify potential prognostic factors affecting prognosis. The multivariable analyses, using a Cox proportional hazard model, included potential predictors with a *p*-value < 0.20 in the univariate analysis. Differences with a two sided *p*-value of <0.05 were regarded as significant. Additionally, to control for possible heterogeneity between the groups with or without ENE in terms of perioperative characteristics, a propensity score matched analysis was performed. Using one to one nearest neighbor matching, a balanced cohort was generated. Comparisons between the matched groups were then made using the paired t-test or Wilcoxon rank sum test for continuous variables and the McNemar test for categorical variables. The SPSS version 22.0 software (IBM, Armonk, NY, USA) was applied for statistical analyses.

## 3. Results

### 3.1. Clinicopathologic Characteristics of Extranodal Extension

The median duration of postoperative surveillance for the 282 patients was 54.3 months (range, 0.7–177.4 months). The patients comprised 179 men (63.5%) and 103 women (36.5%), with a median age of 65.0 years (range, 31.0–83.0 years). Table 1 presents the patients’ clinical characteristics and pathologic findings. Among the 282 patients, there were 199 adenocarcinoma cases, 61 squamous cell carcinoma cases, and 22 cases of cancers with other cell types. ENE was detected in 85 patients (30.1%), and the clinicopathologic characteristics of patients depending on ENE status are shown in Table 1. Significant differences were found in patients’ clinicopathologic characteristics according to the presence of ENE, which was profoundly associated with advanced pathologic findings. The presence of ENE showed an association with T stage (*p* = 0.034), N2 subgroups (*p* < 0.001), and lymphatic invasion (*p* = 0.001). Furthermore, ENE positive patients also had a higher likelihood of undergoing pneumonectomy than those who did not have ENE (10.6% vs. 4.1%, *p* = 0.002).

### 3.2. Prognostic Factors for Disease-Free Survival

The prognostic factors evaluated in the study included age, sex, smoking status, the presence of ENE, preoperative pulmonary function, T stage, nodal status, other pathologic findings, the extent of resection, and adjuvant therapy. The multivariable analysis using a Cox proportional hazard model indicated that old age (*p* < 0.001), T stage (*p* = 0.012), N2 subgroups (*p* = 0.005), and ENE (hazard ratio [HR] 1.629, 95% confidence interval [CI]: 1.161 to 2.284, *p* = 0.005) were independent prognostic factors for DFS (Table 2). The DFS rate at five years was 21.4% for patients who had ENE and 43.4% for patients who did not have ENE (*p* < 0.001, Figure 2).

There were 84 patients in the N2a1 subgroup, 130 patients in the N2a2 subgroup, and 68 patients in the N2b subgroup. Further analyses of the three subgroups were performed. The data presented in Figure 3 show the DFS curves according to the presence of ENE in different N2 subgroups. ENE positive patients had a worse prognosis than patients who did not have ENE across all subgroups (N2a; *p* = 0.253, N2a2; *p* = 0.037, and N2b; *p* = 0.065).

Since there were significant clinicopathologic differences according to the presence of ENE, a propensity score matched analysis was additionally carried out with the goal of determining whether ENE is valuable as an independent prognostic factor for patients with stage III-N2 NSCLC following curative pulmonary resection. Propensity score matching yielded two groups, each of which contained 78 patients. These groups were well balanced, as shown in Table 3. In the matched patients, DFS was significantly different according to the presence of ENE (*p* = 0.001, Figure 4). The DFS rates at five years were 48.0% in the ENE negative and 21.4% in the ENE positive groups, respectively; these results were similar to the nonmatched results.

## 4. Discussion

The principal finding of this study is that ENE showed significant associations with advanced nodal status (*p* < 0.001), lymphatic invasion (*p* = 0.001), and pneumonectomy (*p* = 0.009). These findings are important because the prognostic value of ENE has been undervalued in lung cancer.

The TNM stage of NSCLC is inarguably the prognostic factor with the greatest importance. Based on the eighth edition of the International Association for the Study of Lung Cancer TNM classification, the T stage is classified based on tumor size in 1-cm intervals. The newest edition recommended a more detailed classification system, but N staging remains comparatively simple: N1, N2, and N3 [22]. The N stage can be classified into subgroups (N1a, N1b, N2a1, N2a2, N2b), but this system is not used in clinical practice and this subclassification alone cannot explain the heterogeneity of the N2 group.

ENE is known to be an important feature of the aggressive histopathologic tumor phenotype and has a substantial prognostic influence. ENE positivity in a tumor broadly and negatively impacts recurrence and survival, and numerous studies have demonstrated its prognostic significance for several cancer types, such as breast [7,8,9,10], head and neck [11,12,13], oral cavity [14], colorectal [15,16], gastric [17], cervical [18], and vulvar cancers [19]. In each of the solid cancers, ENE is a risk factor for all cause mortality. The importance of the prognostic implications of ENE was reflected in the latest version of staging for vulvar squamous cell carcinoma [19].

In contrast, the role of ENE and its clinicopathologic implications in NSCLC have not been fully investigated; limited research has reported the prognostic value of ENE [20,24]. In a review of 13 observational studies including 1709 NSCLC patients with LN metastasis, Luchini and colleagues concluded that ENE showed strong associations with recurrence and all cause mortality [20]. They reported that ENE was present at a higher frequency in women, patients who had adenocarcinoma-subtype tumors, and patients with advanced tumor stages; the latter two findings were also observed in our study. However, this study included all N1-3 stages and therefore did not fully evaluate patients’ prognoses. Another systematic review and meta-analysis by Tabatabaei and colleagues [24] analyzed two retrospective and three prospective observational studies including 828 patients diagnosed with NSCLC with LN metastasis. They also provided evidence for an association of ENE with an unfavorable prognosis of NSCLC, as well as with high grade tumors, tumor protein p53 overexpression, lymphatic and vascular invasion. These could be the pathophysiologic causes for the poor outcomes of patients with ENE. However, this review had a statistical limitation, as three studies did not report data on DFS and cancer specific mortality. In addition, their study did not evaluate the association of ENE with clinicopathologic characteristics in N2 subgroups. Nomura and colleagues recently analyzed 168 lung adenocarcinoma patients and concluded that ENE was the most meaningful prognostic factor in N1 and N2 disease [25]. However, in the current study, we analyzed 282 patients and found that ENE status was the most meaningful predictor of survival, regardless of histologic cell type.

Our study identified that clinicopathologic findings in postresection pN2 disease were associated with a poor prognosis, and we showed associations between ENE and advanced nodal stage, lymphatic invasion, and pneumonectomy. Patients with ENE had more advanced pathologic stages and aggressive histopathologic features, which could be explained by the fact that more of these patients underwent pneumonectomy (Table 1 and Table 3). Furthermore, our study addressed the limitations of the above studies by subclassifying the patients by N2 groups into N2a1, N2a2, and N2b subgroups. As shown in Figure 2, our data supported previous studies reporting that ENE presence was associated with an unfavorable prognosis, but further showed that the prognosis worsened incrementally across the N2 groups (Figure 3). Moreover, to our knowledge, no propensity score matching study has yet demonstrated the prognostic value of ENE. After performing propensity score matching, we successfully excluded the confounders and prognostic effects of other invasive histopathologic features. ENE was significantly associated with a poor prognosis and was incrementally associated with an unfavorable prognosis in the matched N2 groups (Figure 4). ENE is therefore a prognostic factor, regardless of TNM stage, and given the unambiguous evidence supporting the prognostic role of ENE herein, we suggest that ENE presence might be included in the next TNM staging.

This retrospective study has some limitations. First, the retrospective study design made it impossible to avoid time-trend and patient selection biases in the treatment for N2 disease. Nonetheless, an unbiased sample was generated through propensity-score matching. Second, this study was conducted in a relatively small cohort at a single center. However, as shown through the analysis above, this study provides data with sufficient statistical power. Third, analyses of epidermal growth factor receptor (*EGFR*) and Kristen-Rous sarcoma virus (*KRAS*) mutations and anaplastic lymphoma kinase (*ALK*) rearrangement were performed only in 162 patients (57.4%), 144 patients (51.1%), and 89 patients (31.6%), respectively. Since the mutational profile information was inadequate for a proper evaluation, we could not include the mutational profile in our analysis.

## 5. Conclusions

In conclusion, among clinical stage III-N2 NSCLC patients, ENE showed a significant association with a poor prognosis, regardless of the TNM stage. Therefore, ENE can be a meaningful prognostic factor in the N2 subgroup, and other adjuvant and multidisciplinary efforts must be made to improve the prognosis of these high risk patients. Moreover, recognizing ENE in the new nodal staging system should be considered.

## Figures and Tables

**Figure 1 jcm-10-03324-f001:**
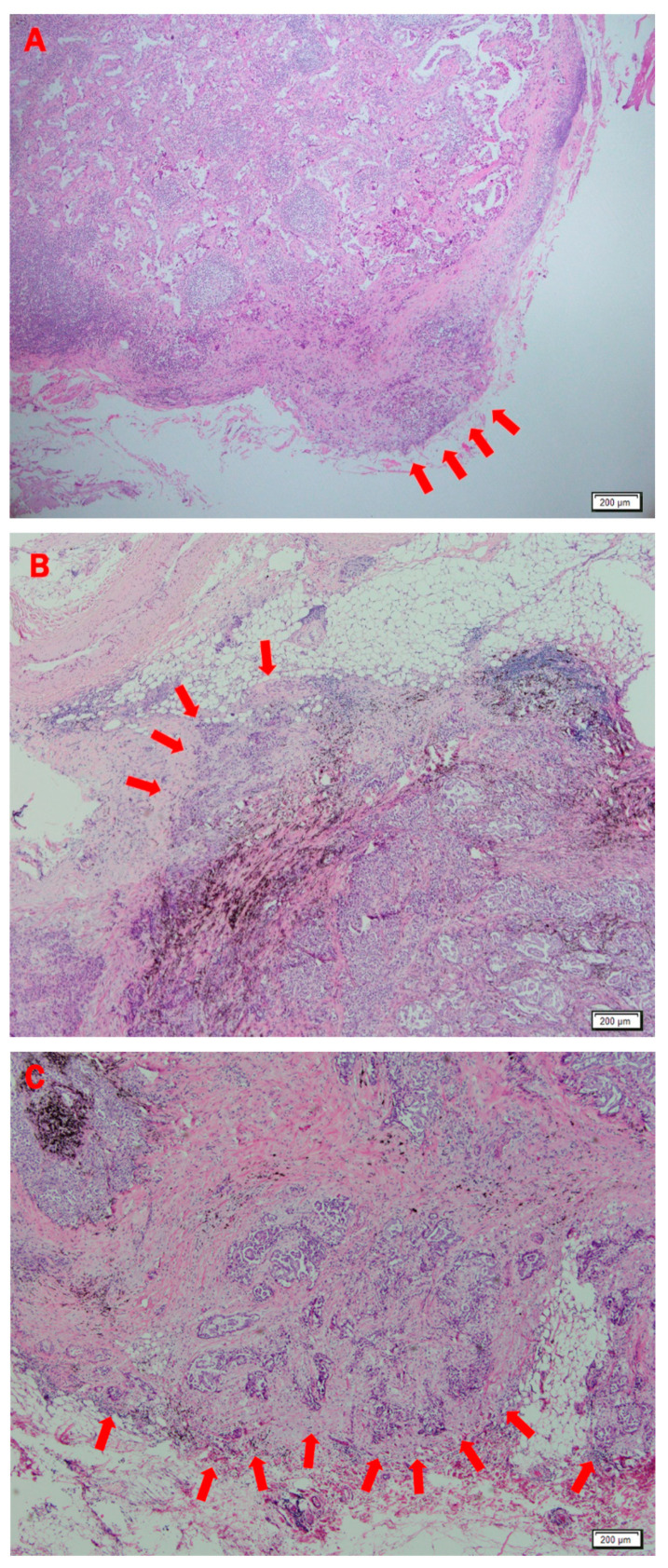
Histologic findings indicating extranodal extension (ENE) in lung adenocarcinoma (hematoxylin and eosin; ×40). Arrows denote ENE with cancer cells extending through the capsule of the lymph node, invading adjacent fatty tissue.

**Figure 2 jcm-10-03324-f002:**
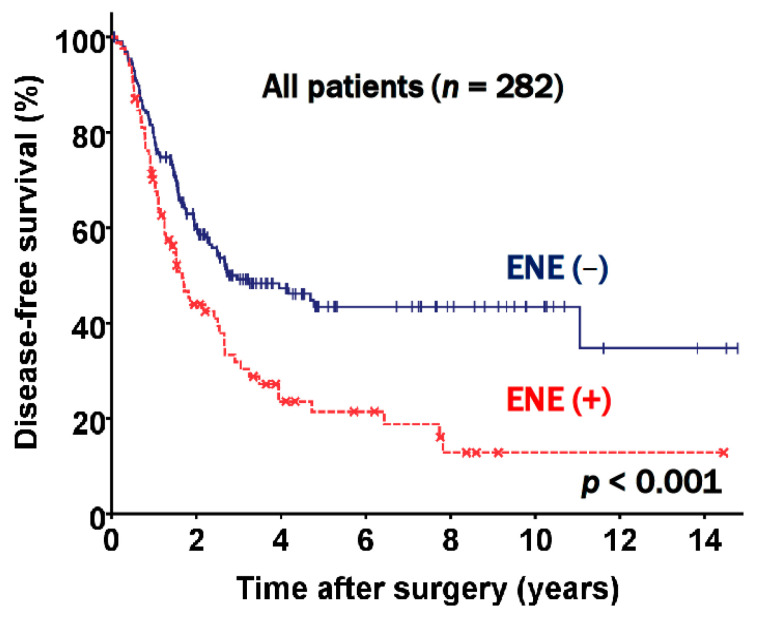
Disease free survival curve according to extranodal extension (ENE) status.

**Figure 3 jcm-10-03324-f003:**
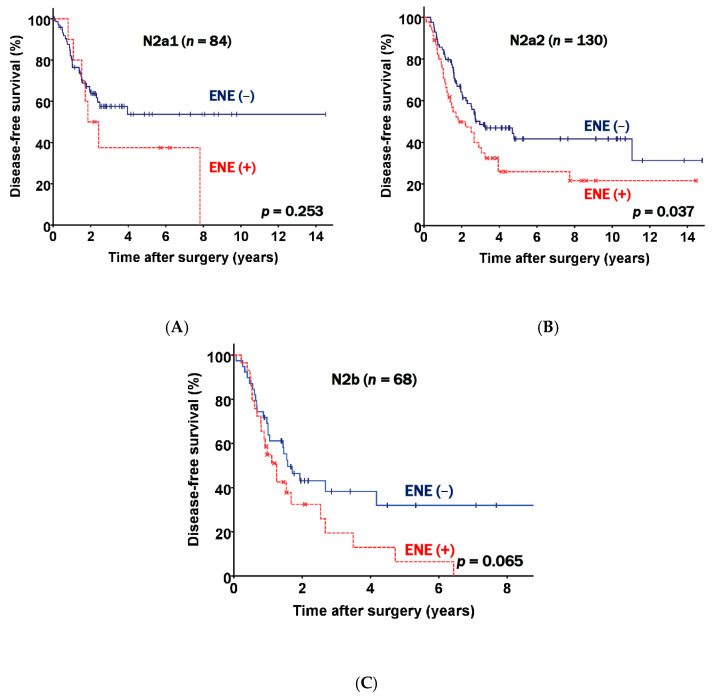
Disease free survival curves classified according to nodal status and extranodal extension (ENE) status. (**A**) N2a1 patients. (**B**) N2a2 patients. (**C**) N2b patients.

**Figure 4 jcm-10-03324-f004:**
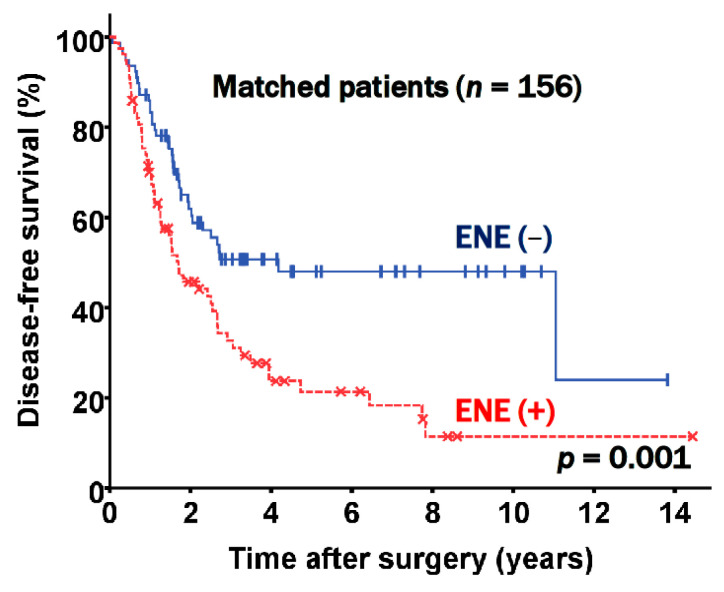
Disease free survival according to extranodal extension (ENE) status in matched patients.

**Table 1 jcm-10-03324-t001:** Patients’ clinical characteristics depending on status of extranodal extension.

	ENE Positive (%) (*n* = 85)	ENE Negative (%) (*n* = 197)	*p*-Value
Age (years)	62.1 ± 10.3	64.5 ± 10.4	0.078
Sex (male)	58 (68.2)	121 (61.4)	0.275
Histology			0.780
Adenocarcinoma	59 (69.4)	140 (71.1)	
Squamous cell carcinoma	19 (22.3)	42 (21.3)	
Other cell types	7 (8.3)	15 (7.6)	
FEV1 (%)	99.0 ± 23.5	101.1 ± 19.2	0.455
DLCO (%)	99.9 ± 20.4	102 ± 21.3	0.421
Never smoker	31 (36.5)	87 (44.2)	0.230
T stage			0.034
T1	12 (14.1)	42 (21.3)	
T2	51 (60.0)	125 (63.4)	
T3	12 (14.1)	23 (11.7)	
T4	10 (11.8)	7 (3.60)	
N stage			<0.001
N2a1	10 (11.8)	74 (37.6)	
N2a2	46 (54.1)	84 (42.6)	
N2b	29 (34.1)	39 (19.8)	
Extent of operation			0.002
Lobectomy	76 (89.4)	193 (98.0)	
Pneumonectomy	9 (10.6)	4 (2.0)	
Adjuvant treatment	75 (88.2)	174 (88.3)	0.983
Visceral pleural invasion (+)	51 (60.0)	95 (48.2)	0.069
Vascular invasion (+)	63 (74.1)	129 (65.5)	0.153
Lymphatic invasion (+)	49 (57.6)	72 (36.5)	0.001
Perineural invasion (+)	13 (15.3)	17 (8.60)	0.096

Data are presented as the number (%) or mean ± standard deviation. ENE: extranodal extension; DLCO: carbon monoxide diffusing capacity; FEV1: forced expiratory volume in 1 s.

**Table 2 jcm-10-03324-t002:** Prognostic factors affecting disease free survival.

Variables	Univariate Analysis	Multivariable Analysis	
	*p*-Value	HR (95% CI)	*p*-Value
Age	0.007	1.031 (1.014–1.049)	<0.001
Sex (male)	0.275		
Never smoker	0.559		
Preoperative FEV1 (%)	0.380		
Preoperative DLCO (%)	0.529		
Presence of ENE	<0.001	1.629 (1.161–2.284)	0.005
T stages (T1)	<0.001		0.012
T2		1.392 (0.799–2.424)	
T3		2.526 (1.338–4.770)	
T4		2.158 (0.974–4.784)	
N stages (N2a1)	<0.001		0.005
N2a2		0.995 (0.660–1.501)	
N2b		1.784 (0.990–2.105)	
Visceral pleural invasion (+)	0.001		
Vascular invasion (+)	0.306		
Lymphatic invasion (+)	0.497		
Perineural invasion (+)	0.280		
Histology (adenocarcinoma)	0.695		
Operation type (pneumonectomy)	0.622		
Adjuvant treatment (+)	0.189		

Data are presented as the number (%) or mean ± standard deviation. CI: confidence interval; ENE: extranodal extension; DLCO: carbon monoxide diffusing capacity; FEV1: forced expiratory volume in 1 s.

**Table 3 jcm-10-03324-t003:** Clinical characteristics of patients depending on the status of extranodal extension after propensity score matching.

	ENE Positive (%) (*n* = 78)	ENE Negative (%) (*n* = 78)	*p*-Value
Age (years)	62.6 ± 10.3	61.7 ± 11.7	0.593
Sex (male)	53 (67.9)	53 (67.9)	1.000
Histology (adenocarcinoma)	54 (69.2)	49 (61.5)	0.564
FEV1 (%)	99.7 ± 24.1	99.0 ± 21.1	0.859
DLCO (%)	100.5 ± 20.5	98.9 ± 21.4	0.642
Never smoker	29 (37.2)	28 (35.9)	1.000
T stage			0.278
T1	11 (14.1)	12 (15.4)	
T2	48 (61.5)	52 (66.7)	
T3	11 (14.1)	12 (15.4)	
T4	8 (10.3)	2 (2.6)	
N stage			0.968
N2a1	9 (11.5)	10 (12.8)	
N2a2	42 (53.8)	41 (52.6)	
N2b	27 (34.6)	27 (34.6)	
Extent of operation			0.534
Lobectomy	71 (91.0)	74 (94.9)	
Pneumonectomy	7 (9.00)	4 (5.10)	
Adjuvant treatment	68 (87.2)	72 (92.3)	0.667
Visceral pleural invasion (+)	46 (59.0)	41 (52.6)	0.519
Vascular invasion (+)	56 (71.8)	50 (64.1)	0.391
Lymphatic invasion (+)	42 (53.8)	38 (48.7)	0.631
Perineural invasion (+)	9 (11.5)	10 (12.8)	1.000

Data are presented as the number (%) or mean ± standard deviation. ENE: extranodal extension; DLCO: carbon monoxide diffusing capacity; FEV1: forced expiratory volume in 1 s.

## Data Availability

Data available on request due to restrictions such as privacy or ethics.

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
