# Peer review of "Prognostic Significance of the Extranodal Extension of Regional Lymph Nodes in Stage III-N2 Non-Small-Cell Lung Cancer after Curative Resection"

_jcm, 2021, doi:10.3390/jcm10153324_

Round 1
Reviewer 1 Report
Chia-Hui Shih and co-authors described the prognostic significance of extranodal extension (ENE) in patients with stage III-N2 non-small-cell lung cancer (NSCLC) who underwent curative surgery. They performed an extensive work analyzing a big cohort of patients. They demonstrated that, among the 282 patients, ENE was detected in 30.1% of the patients and it was associated with advanced T stage, advanced nodal status, lymphatic invasion and pneumonectomy. In addition, they found that old age, advanced T stage, advanced nodal status and the presence of ENE were significant independent prognostic factors for DFS. Moreover, the 5-year DFS rate was lower for patients with ENE and higher for patients without ENE. They concluded that ENE associated with TNM staging may represent an important prognostic factor for patients with stage III-N2 NSCLC after curative resection.
The paper is well written and flows well and logically.
Main comments:
- the title leads to NSCLC, however, only adenocarcinomas are considered for the analysis. Please, change the title or include patients with squamous carcinomas
-is there any correlation between the mutational status of the tumors included in the study and the presence of ENE?
-please, provide several H&E images
Minor comment:
-please, add the scale bar in Figure 1
Author Response
Reviewer #1 Chia-Hui Shih and co-authors described the prognostic significance of extranodal extension (ENE) in patients with stage III-N2 non-small-cell lung cancer (NSCLC) who underwent curative surgery. They performed an extensive work analyzing a big cohort of patients. They demonstrated that, among the 282 patients, ENE was detected in 30.1% of the patients and it was associated with advanced T stage, advanced nodal status, lymphatic invasion and pneumonectomy. In addition, they found that old age, advanced T stage, advanced nodal status and the presence of ENE were significant independent prognostic factors for DFS. Moreover, the 5-year DFS rate was lower for patients with ENE and higher for patients without ENE. They concluded that ENE associated with TNM staging may represent an important prognostic factor for patients with stage III-N2 NSCLC after curative resection.
The paper is well written and flows well and logically.
<Main comments>
Point 1: The title leads to NSCLC, however, only adenocarcinomas are considered for the analysis. Please, change the title or include patients with squamous carcinomas
Response 1: Thank you for your comment and we are very sorry for the confusion we made. In this study, we included patients with stage IIIA-N2 disease, regardless of histologic types. In other words, we included not only adenocarcinoma but also squamous cell carcinoma, and other cell type tumors. As you can see from Table 1 (original manuscript), of the 282 patients evaluated, there were 199 patients (70.6%) with lung adenocarcinoma. As reviewer suggested, we clearly remark the detailed histologic cell type of patients in the Result Section (“Among the 282 patients, there were 199 adenocarcinoma cases, 61 squamous cell car-cinoma cases, and 22 cases of cancers with other cell types.”) and Table 1.
Point 2: Is there any correlation between the mutational status of the tumors included in the study and the presence of ENE?
Response 2: We really appreciate your thorough review and advice. In this study, we analyzed patients with stage IIIA-N2 NSCLC disease, who underwent curative resection between 2005 and 2018, and our investigation during the retrospective review showed lack of mutation profiles examined in these patients. Analyses of EGFR and KRAS mutations and ALK rearrangement were performed only in 162 patients (57.4%), 144 patients (51.1%), and 89 patients (31.6%), respectively. Since the mutational profile information was inadequate for a proper evaluation, we could not include the mutational profile in our analysis. Also, as reviewer suggested, we clearly remarked the limitation of this study in the “last paragraph of Discussion Section”. However, as you have mentioned, this will be a topic of future investigation. We appreciate your outlook.
Point 3: Please, provide several H&E images
Response 3: We really thank you for your constructive and helpful remark. As reviewer has suggested, we additionally included two more histologic figures (Figure 1 A, B, and C) that shows ENE. The slides have scale bar and have been evaluated by our dedicated pathologist to improve the paper’s integrity.
<Minor comment>
Point 4: Please, add the scale bar in Figure 1
Response 4: Thank you for your thoughtful comment. As you have commented, we included scale bar in the histologic figures (Figure 1 A, B, and C).
Reviewer 2 Report
The paper delves into the relationship between ENE status, clinicopathological characteristics in Stage III-N2 lung cancer patients. The authors have made a good effort at presenting their findings in a scientifically sound manner. Though the sample size is relatively smaller than similar studies, the analysis brings to light the importance of ENE status in Stage III N2 (including subtypes) cancer patients.
Author Response
Point 1: The paper delves into the relationship between ENE status, clinicopathological characteristics in Stage III-N2 lung cancer patients. The authors have made a good effort at presenting their findings in a scientifically sound manner. Though the sample size is relatively smaller than similar studies, the analysis brings to light the importance of ENE status in Stage III N2 (including subtypes) cancer patients.
Response 1: Thank you for your positive review and kind words on our paper.
Reviewer 3 Report
Thank you for submitting this article. I was pleased to receive it as a reviewer.
I have the following questions for you, which I believe, need to be addressed before publication:
First, the paper should be written according to the STROBE (Strengthening the reporting of observational studies in epidemiology) [www.strobe-statement.org]. A STROBE checklist should also be added [https://www.strobe-statement.org/fileadmin/Strobe/uploads/checklists/STROBE_checklist_v4_combined.doc].
The statistical analysis should be written according to the recently published guidelines (Hickey GL, Dunning J, Seifert B, Sodeck G, Carr MJ, Beyersdorf F on behalf of the EJCTS and ICVTS Editorial Committees Editor's Choice: Statistical and data reporting guidelines for the European Journal of Cardio-Thoracic Surgery and the Interactive CardioVascular and Thoracic Surgery. Eur J Cardiothorac Surg 2015;48:180-93).
The limitations section should be improved with a better discussion.
In addition, the discussion should be improved with a better search of the literature.
About minor points, there are grammars and typos errors in the text. Please thoroughly check the article.
Good luck with your article, and thanks again for submitting it.
Author Response
Thank you for submitting this article. I was pleased to receive it as a reviewer. I have the following questions for you, which I believe, need to be addressed before publication:
Point 1: First, the paper should be written according to the STROBE (Strengthening the reporting of observational studies in epidemiology) [www.strobe-statement.org]. A STROBE checklist should also be added [https://www.strobe-statement.org/fileadmin/Strobe/uploads/checklists/
STROBE_checklist_v4_combined.doc].
Response 1: Thank you for your kind review comment. As reviewer has commented, we evaluated whether our paper abides by STROBE criteria, and we find that this study followed the guidelines well. We have also attached the STROBE check list, as reviewer suggested.
Point 2: The statistical analysis should be written according to the recently published guidelines (Hickey GL, Dunning J, Seifert B, Sodeck G, Carr MJ, Beyersdorf F on behalf of the EJCTS and ICVTS Editorial Committees Editor's Choice: Statistical and data reporting guidelines for the European Journal of Cardio-Thoracic Surgery and the Interactive CardioVascular and Thoracic Surgery. Eur J Cardiothorac Surg 2015;48:180-93).
Response 2: Thank you for your kind and thoughtful comment. As we have clarified in the Acknowledgment section, we sought statistical advice from Jung Hoon Lee, a specialist in biomedical informatics from Seoul National University Biomedical Informatics. We believe that statistical analysis has been evaluated as according to the guideline.
Point 3: The limitations section should be improved with a better discussion.
Response 3: Thank you for your constructive and thoughtful comment. We recognize that our study lacks in analysis of the mutational profile as can be seen from Comment 2 of Reviewer #1, and we clearly addressed the content in the Limitations Section.
Point 4: In addition, the discussion should be improved with a better search of the literature.
Response 4: Thank you for your constructive comment. Thanks to your comment, we were able to include additional evidence, which seems to have improved the quality of our paper. We included the most recent article that is very pertinent to our paper, which was published just last year. In discussion section, we added the following line: “Nomura and colleagues recently analyzed 168 lung adenocarcinoma patients and concluded that ENE was the most meaningful prognostic factor in N1 and N2 disease [25]. However, in the current study, we analyzed 282 patients and found that ENE status was the most meaningful predictor of survival, regardless of histologic cell type.”, and added in Reference List.
Point 5: About minor points, there are grammars and typos errors in the text. Please thoroughly check the article.
Response 5: Thank you for your kind comment. We have checked grammar and errors with professional English Service again and have updated the changes accordingly.
Round 2
Reviewer 3 Report
A statement should be added regarding the STROBE (e.g. We retrospectively reviewed, according to the STROBE (Strengthening the reporting of observational studies In epidemiology) statement, our experience on [...]. A STROBE checklist can be found in the Supplemental Material.).
The reference to STROBE should be added.
Author Response
Response to Reviewer 3 Comments
Point 1: A statement should be added regarding the STROBE (e.g. We retrospectively reviewed, according to the STROBE (Strengthening the reporting of observational studies In epidemiology) statement, our experience on [...]. A STROBE checklist can be found in the Supplemental Material.). The reference to STROBE should be added.
Response 1: We really appreciate your thorough review and advice. As reviewer suggested, we clearly remarked in the manuscript that this study was written according to the STROBE criteria, and also added relevant references (www.strobe-statement.org, and Reference Number #23: Vandenbroucke JP, von Elm E, Altman DG, Gøtzsche PC, Mulrow CD, Pocock SJ et al. Strengthening the Reporting of Observational Studies in Epidemiology (STROBE): explanation and elaboration. Ann Intern Med. 2007;147:W163-94). A STROBE checklist, which was previously attached, was re-attached as a “Supplementary Material (S1)”. The order of the References list was also modified, accordingly.